# Biogenic Silica and Organic Carbon Records in Zhoushan Coastal Sea over the Past One Hundred Years and Their Environmental Indications

**DOI:** 10.3390/ijerph17113890

**Published:** 2020-05-30

**Authors:** Hao Xu, Shangwei Jiang, Jialin Li, Ruiliang Pu, Jia Wang, Wanghai Jin, Longbin Sha, Dongling Li

**Affiliations:** 1Department of Geography and Spatial Information Techniques, Ningbo University, Ningbo 315211, China; 176000005@nbu.edu.cn (S.J.); lijialin@nbu.edu.cn (J.L.); 146331802@nbu.edu.cn (J.W.); 176000007@nbu.edu.cn (W.J.); shalongbin@nbu.edu.cn (L.S.); lidongling@nbu.edu.cn (D.L.); 2Institute of East China Sea, Ningbo University, Ningbo 315211, China; 3School of Geosciences, University of South Florida, Tampa, FL 33620, USA; rpu@usf.edu

**Keywords:** Zhoushan coastal sea, sedimentary record, biogenic silica, organic carbon, provenance

## Abstract

The influence of terrestrial and marine input has dramatically changed eutrophication in coastal seas over the past 100 years. In this study, Zhoushan coastal sea (ZCS) is taken as a study area. We studied ZCS as it is a sink of the temporal and spatial variation of primary productivity, dominant species of algae, and the variation of provenance in this area over the past 100 years. We performed analysis using three sediment cores and the carbon and silicon deposition records. The analysis results demonstrate that: (1) The primary productivity in the northern area of the ZCS close to the Yangtze Estuary was the highest comparatively, but it declined slightly before 2010. The primary productivity in the southern area had an increasing trend over the past 100 years. The value of total organic carbon (TOC) in the northern area was relatively high, with an average value of 0.532% over the past 100 years, with a decreasing trend in recent years. On the contrary, TOC in the southern area was relatively low, but it was increased dramatically after 1995. (2) Diatom might play an important role in the variations. The biogenic silica (BSi) and TOC in the northern area showed a synchronous declining trend, while the BSi/TOC ratio did not change significantly. This indicates the algae population structure in this area was relatively stable over the past 100 years. The BSi/TOC ratio decreased continuously in the southern area, indicating that the dominance of diatoms was decreasing continuously. (3) The variation of diatom dominance in this area might have a great relationship with the change of nutrients’ provenance. A mean value of stable carbon isotope (δ^13^C) in the north of Zhoushan was −23.46‰, indicating that the terrestrial-source input was the highest. (4) The change of provenance in the study area was quite different. This illustrates that the terrestrial input impacted the largest area of ZCS while marine input became dominant in the offshore area.

## 1. Introduction

The Zhoushan Coastal Sea (ZCS) is located in the south of the Yangtze Estuary, west of Hangzhou Bay and near the Zhoushan Islands and Ningbo City. The marine ecological environment in the ZCS is significantly influenced by the Yangtze River. It has been proven that there was a serious eutrophication in the ZCS during the past four decades due to human activities in the drainage basin and artificial aquaculture activities in offshore areas [1,2,3,4]. The frequency of occurrence of red tide in this area has also significantly increased since 1980s, accompanied with an increasing trend of harmful algal blooming [5,6,7,8,9]. Although many studies have proven that illumination, water temperature, and nutrients are important factors that influence algae growth [10,11], it is necessary to study whether overloading nutrients and changes of nutrient ratio are a key cause of changes in the dominant algal species of red tides in the ZCS during the past four decades associated with changes of illumination and water temperature [12,13].

Currently, it is widely accepted that changes of nutrient ratio in the ZCS have been influenced by nutrients fluxes from the Yangtze River to the sea. However, the role of marine sourced nutrients in harmful algal blooming of the ZCS is still worthy of a further study, as this area is also influenced by the upwelling which may bring numerous marine sourced nutrients. The variation of nutrients and their sources in the overlying water can be kept and recorded well by core sediments in offshore areas [14,15,16]. By testing and checking parameters, such as total organic carbon (TOC), biogenic silica (BSi), and stable carbon isotope(δ^13^C), in core sediments and considering age frame, the sedimentary records of organic matters and their sources in the overlying water could be reconstructed. This method has been widely applied in worldwide mega estuaries, such as the Yangtze Estuary, Mississippi Estuary, Nile Estuary, Pearl River Estuary, and so on [17,18,19,20]. Researchers reconstructed records of the algae community’s variation during red tides in the East China Sea over the past century by using core sediments in offshore areas [21,22,23]. Those studies mainly focused on the sea area in the Yangtze Estuary and inner continental shelf of the East China Sea. According to the series of core test results in offshore areas in the Yangtze Estuary, the degree of dominance of dinoflagellate community presented an increasing trend in recent years, which was influenced by terrestrial source inputs from the Yangtze River [24,25]. Xing (2016) pointed out an increasing trend of degree of dominance of diatom in the South Zhejiang Offshore areas by using changes of ratio of brassicasterol to dinoflagellates sterol [26]. This trend might be related with marine source inputs, such as strengthened upwelling. This reflected that material sources played an important role in changes of local eutrophication.

The ZCS is located between the Yangtze Estuary and the South Zhejiang Offshore and is a sensitive and interactive sea area between marine sources and terrestrial sources. Influences of marine source and terrestrial source inputs on ecological environmental changes in the area during the past century could be reproduced by reconstructing the algae community structure and discussing its influencing factors. Research results can further provide directions for effective management of coastal environment. Therefore, in this study, sediments in three cores were first collected, and the main objectives include (1) analyzing sedimentary records of biogenic elements, (2) testing the sedimentary record of primary productivity and dominant species variation of algae in the ZCS over the past 100 years, and (3) revealing and assessing the spatiotemporal variations of provenance of biogenic elements in this area over time.

## 2. Materials and Methods

### 2.1. Collection of Samples

Cores at three stations A7-1, A10-4, and A11-3 (Figure 1) in the area were collected by a gravity sampler in March 2016, and then they were kept in refrigerator under −20 °C before being tested. Core A7-1 locates at 30.56 N, 122.50 E with a water depth about 20 m and a length of 170 cm. This core is nearby the north of Zhoushan Islands and in the northern coastal area of the ZCS. Core A10-4 is located at 29.42 N, 122.75 E with a water depth about 50 m and a length of 145 cm. This core presents in the southeast offshore area of the ZCS. Core A11-3 locates at 29.06 N, 122.50 E with a water depth about 40 m and a length of 140 cm, and is in the southern offshore area of the ZCS. All samples were collected by the State Key Laboratory of Estuarine and Coastal Research (East China Normal University) in May 2016. Each core was sampled at an interval of 2 cm and then stored in poly vinyl chloride sealed bags independently. For each core, the excess ^210^Pb(^210^Pb_ex_), biogenic silica (BSi), total organic carbon (TOC) and stable carbon isotope (δ^13^C) were tested. A total of 45 samples were tested for ^210^Pb_ex_, and a total of 225 samples for BSi, TOC, and δ^13^C (Table 1), respectively.

### 2.2. Methods

#### 2.2.1. Pb_ex_ Test

The measurement of ^210^Pb_ex_ was conducted by using DeMaster (1981) [27]. About 10 g wet sample was weighed and dried under 105 °C. The dry sample was weighed again to calculate the water content. Then, the sample was grinded and screened by a 100-mesh sieve to eliminate plant roots. About 3 g of the grinded and screened sample was weighed by a balance and put into a plastic tube that was then sealed up. After waiting for a radioactive energy balance for 20 days, a deposition rate was tested by a Gamma Spectrometer (HPGe GWL-120-15 EG&G/ORTEC, Oak Ridge, TN, USA). The measurement time was set about 4 × 10^4^ s. Total ^210^Pb activity (^210^Pb_t_) was determined by measuring the 46.5 keV g-ray peak, and activities of supported ^210^Pb (^210^Pb_s_) were determined using the g-ray peaks of ^214^Pb (351.9 keV) and ^214^Bi (609.3 and 1120.3 keV). Finally, the excess ^210^Pb was calculated by using the formula: ^210^Pb_ex_ = ^210^Pb_t_ − ^210^Pb_s_ s. The sedimentation rate of the present study was based on a constant rate supply model of ^210^Pb_ex_ [27,28].

The deposition rate was calculated by using the formula: r = λ × Z/(lnAz − lnA_0_) [29], where, r is the deposition rate; λ is the constant 0.03108 (i.e., ln2/22.3); Z is the depth of sample in the core; Az is the ^210^Pb_ex_ value of the sample at depth Z; A_0_ is the ^210^Pb_ex_ value of the surface sample (i.e., depth at 0). The CIC model of ^210^Pb_ex_ [29] was used to calculate ^210^Pb_ex_ values at different depths in the core.

#### 2.2.2. Biogenic Silicon (BSi) Test

A core sample was dried under 60 °C and grinded. Then, a 110–130 mg sample was put in a 50 mL polypropylene centrifuge tube, in which 5 mL 10% H_2_O_2_ and 5 mL 1.0 mol/L HCl were added. The mixture was then processed by an ultrasonic oscillation for 30 min and then centrifuging at the rate of 4000 r/min for 5 min. After supernatant liquid was eliminated, the remainder was supplemented with 20 mL deionized water to eliminate residual HCl and H_2_O_2_. This process repeated until the pH of supernatant liquid close to 7. Next, the sample was dried overnight in an oven under 60 °C. Then, a 40 mL Na_2_CO_3_ solution was added into the centrifuging tube and heated in a water bath at 90 °C. Five hours later, the mixture was centrifuged at the rate of 4000 r/min for 5 min and 0.25 mL supernatant liquid was mixed with 17.5 mL acid ammonium molybdate completely. Another 7.5 mL reducing agent was added and the mixture was oscillated completely. After 2 h of color development, absorbance of the sample was tested at 812 nm wavelength with a Visible Spectrophotometer (7200 Unico, Shanghai, China). The test result was converted into a BSi ratio. 

#### 2.2.3. TOC and δ^13^C Tests

A vacuum freezing and drying sample (about 0.5 g) which was screened by an 80-mesh sieve was put into a centrifuging tube, in which 5.0 mL 10% HCl was added. The mixture was stirred continuously. Then, deionized water was added into the centrifuging tube for cleaning and centrifuging process. This process was repeated until the pH of supernatant liquid was close to 7, before the supernatant liquid was eliminated. The rest sample was dried overnight in an oven under 60 °C. About 50–60 mg of the sample was weighed by a balance and wrapped in a small stannum box. Then, the sample was crushed into sheets. TOC was measured by an element analyzer vario MACRO cube [30]. The detection limit of carbon was 8 × 10^6^ g, the precision of the instrument was <0.01%, and the recovery efficiency was >99.5% [30]. Numerical value of δ^13^C was measured by a DELTA plusXP stable isotope mass spectrometer (ThermoFinnigan, San Jose, CA, USA) [31]. Stable isotopic ratios of δ^13^C were determined as follows: δ^13^C = (R_sample_/R_standard_ − 1) × 1000, where R_sample_ is the test result of ratio ^13^C/^12^C, and R_standard_ is expressed relative to the Vienna Pee Dee Belemnite (V-PDB) standard. The analytical precision was ±0.2‰ for δ^13^C, based on the analysis of the standards [31].

## 3. Results and Analysis

### 3.1. Determination of Chronological Framework

Test results of ^210^Pb_ex_ are shown in Figure 2. In Figure 2, y is depth and x is ln (^210^Pb_ex_). The gradient (i.e., (lnA_0_ − lnAz)/Z) could be changed here: r = λ/gradient [29]. The gradient at A7-1 was −41.462, so the annual average deposition rate was estimated to be at 1.560 cm. The gradient at A10-4 was −33.766, so the annual average deposition rate was estimated at 1.851 cm. The gradient at A11-3 was −43.939, so the annual average deposition rate was estimated at 1.487 cm.

According to the deposition rate, core A7-1 recorded the variation from 1905 to 2015; core A10-4 recorded from 1940 to 2015; and core A11-3 recorded from 1920 to 2015. The variation of primary productivity and provenance over the past 100 years could be revealed. Furthermore, different depths of each core could be transferred into different ages of each core, shown in Figure 3, Figure 4, Figure 5 and Figure 6, according to their corresponding deposition rates.

### 3.2. Primary Productivity Records

Figure 3 presents test results of TOC of three cores, the error of the results less than 0.01%. For core A7-1, TOC value declined gradually with an average value of 0.532%. The maximum and minimum of the TOC were 0.703% in 1930 and 0.303% in 2000, respectively. For core A10-4, TOC value presented an increasing trend. From 1940 to 1995, the TOC value increased slowly. In this period, the maximum and minimum were 0.445% in 1990 and 0.187% in 1955, showing an average value of 0.326%. From 1996 to 2015, TOC increased dramatically. In this period, the maximum and minimum were 0.689% in 2014 and 0.321% in 2000, with an average value of 0.486%. Per core A11-3, TOC value increased quickly during 1920–1950. The minimum and maximum of the TOC were 0.343% in 1925 and 0.649% in 1950, with an average value of 0.519%. However, the TOC value became relatively stable after 1950. The maximum and minimum of TOC were 0.704% in 1970 and 0.5434% in 2008, showing an average value of 0.630%.

During the past 100 years, the TOC values for most samples in cores A7-1 and A11-3 were around 0.6%, while TOC values for most samples in core A10-4 were only 0.4%, indicating that core A10-4 had a relatively low sediment of total organic matters. This might be due to the deepest location of core A10-4 among the three cores. However, the dramatic increase of TOC value in core A10-4 after 1995 might reveal that the deposited model of TOC has changed recently.

Test results of BSi of the three cores are shown in Figure 4. For core A7-1, BSi value was decreasing continuously. It declined slightly from 1905 to 1985. The maximum was 0.782% in 1945 and the minimum was 0.447% in 1925, with an average of 0.582%. However, BSi of samples from core A7-1 declined quickly from 1985 to 2015. The maximum was 0.637% in 1990 and the minimum was 0.369% in 2008, with an average of 0.446%. For core A10-4, the BSi value was increasing continuously, manifested by a small fluctuation during 1940–1980 and a great fluctuation during 1981–2015. The maximum and minimum of the BSi were 0.754% in 2000 and 0.371% in 1945, showing an average value of 0.507%. For core A11-3, the BSi generally increased slightly during 1920–1995. The maximum and minimum of the BSi were 0.616% in 1960 and 0.421% in 1930, with an average value of 0.513%. Then, it dropped dramatically after 1996. Its maximum and minimum were 0.568% in 2010 and 0.426% in 2005, showing an average value of 0.484%.

Generally speaking, the TOC and BSi in A7-1 both declined slightly over the past 100 years, while the TOC and BSi in A10-4 both increased. This might indicate that the primary productivity in the northern Zhoushan costal sea (ZCS) has not increased but it in the southern ZCS has increased. The TOC in A11-3 has increased while the BSi has declined since 2000, which might indicate that the advantage of diatom has declined in this core since 2000.

### 3.3. Provenance Analysis

Test results of δ^13^C of the three cores are shown in Figure 5. All the three cores were with marine-terrestrial mixed source. For core A7-1, value of δ^13^C was relatively high and stable during 1905–1970. The maximum and minimum numerical values of δ^13^C were −23.00‰ in 1940 and −23.82‰ in 1922, showing an average value of −23.34‰. During 1971–1985, the value of δ^13^C dropped significantly compared to that during 1905–1970. At this stage, the value of δ^13^C presented an increasing trend. The maximum and minimum values of δ^13^C were −23.33‰ in 1971 and −24.17‰ in 1985, with an average of −23.77‰. During 1986–2015, the value of δ^13^C presented a decreasing trend, indicating the increasing terrestrial organic input. The maximum and minimum values of δ^13^C were −23.29‰ in 1986 and −23.91‰ in 2014, with an average of –23.57‰. For core A10-4, the value of δ^13^C increased gradually from 1940 to 1995, indicating the increasing marine input. The maximum and minimum values of δ^13^C were −21.99‰ in 1948 and −23.71‰ in 1970, showing an average of −22.63‰. However, the value of δ^13^C decreased gradually from 1996 to 2015, indicating the increasing terrestrial input too. The maximum and minimum values of δ^13^C were −21.71‰ in 1996 and −22.57‰ in 2009, with an average of −22.24‰. For core A11-3, the value of δ^13^C decreased generally. The maximum and minimum values of δ^13^C were −22.36‰ in 1936 and –24.05‰ in 1986, with an average of −23.01‰.

## 4. Discussion

### 4.1. Variation of Dominace of Diatom in Phytoplankton

BSi represents the productivity of diatom while TOC indicates the total productivity of phytoplankton. The ratio of BSi/TOC can illustrate the dominance of diatom [30]. The results of BSi/TOC extracted from the three cores are shown in Figure 6. For core A7-1, the gradient of BSi/TOC was 0.0003, which indicates that the temporal variation of BSi/TOC was quite stable. The maximum and minimum values were 1.572 in 1935 and 0.723 in 2000, showing an average value of 1.066. This suggests that the dominance of diatom in this area was not changed dramatically over the past 100 years. For core A10-4, the BSi/TOC fluctuated greatly, and it was generally increasing continuously during 1940–1995. The gradient in this period was about 0.0024, which indicates an increasing trend. The maximum and minimum values were 2.312 in 1955 and 0.963 in 1950, showing an average value of 1.484. However, during 1996–2015, it dropped greatly compared to that during 1940–1995. The gradient during 1996–2015 was about −0.0453, indicating a dramatically decrease. During the period, the maximum and minimum values were 1.816 in 2000 and 0.868 in 2015, showing an average value of 1.307. This reflects that the dominance of diatom in this area decreased significantly. For core A11-3, the gradient was −0.0023, which indicates BSi/TOC decreasing slightly. The maximum and minimum values of BSi/TOC were 1.462 in 1930 and 0.673 in 2010, showing an average value of 0.866.

Among the three cores, BSi/TOC at A10-4 was the highest, showing the diatom taking a greater advantage than that in A7-1 and A11-3. However, this advantage at A10-4 has declined since 1986. And BSi/TOC in A11-3 has also decreased slightly. Generally, the dominance of diatom at the three cores has declined.

In this study area, the dominance of diatom showed evident temporal and spatial variations. In the northern area (A7-1), near to Yangtze Estuary, both TOC and BSi have declined continuously, and the BSi/TOC fluctuated slightly. This reflected that the primary productivity in the area continued to decline and the dominance of diatom has not changed dramatically. In the southeastern area (A10-4), both TOC and BSi increased continuously and the value of BSi/TOC increased before 1995, which proved the continuous growth both in primary productivity and the dominance of diatom. However, the growth rate of TOC was significantly higher than that of BSi after 1995, resulting in a quick reduction of BSi/TOC. This means that the primary productivity increased continuously, and the dominance of diatom declined dramatically. In the southern area (A11-3), TOC continued to increase, and it increased greatly before 1950, but the growth rate decreased significantly after 1950. The BSi presented an increasing trend before 1995, but it turned to a reduction trend after 1995. The value of BSi/TOC in this area continued to decrease, which reflected an increasing total productivity, but a decreasing productivity of diatom and decreasing dominance of the diatom. To sum up, the degree of dominance of diatom in the northern area increased. More specifically, it increased firstly and then decreased in the southeastern area, while it decreased continuously in the southern area.

As we summarized above, the dominance of diatom varied in this coastal sea. This was influenced by illumination, water temperature and nutrients ratio significantly. In this study area, water temperature changed slightly and the influence of illumination on algae was small. Yang (2013) carried out a multiple linear regression analysis and found that temperature contributed only 6.8% of interannual variation of plankton in the study area based on observed algal data during the past 20 years [32]. Han (2009) observed that temperature contributed about 9% of plankton growth in the Yangtze Estuary and its adjacent sea areas [33]. Song (2014) analyzed a direct correlation between the variation of Chlorophyll content and total suspended solid (TSS) based on a generalized additive model (GAM) and believed that illumination condition was a decisive factor of temporal and spatial distribution of plankton in sea areas [13]. Zhou et al (2017) simulated succession of algal communities and proved that temperature and illumination affected algal succession slightly [34]. In summary, the previous studies indicate that nutrient is an important factor that influences the dominance of diatom in the ZCS area.

The northern area in the ZCS is close to the Yangtze Estuary. There are overload nutrients from the Yangtze River to seas due to intensive human activities in the river basin. Tang (2009) pointed that annual average concentration of dissolved inorganic nitrogen (DIN) at Yangtze Estuary generally increased continuously in the past 50 years and it increased from 4 μmol/L in 1960s to 27 μmol/L in 2009 [35]. The annual average concentration of P also presented an increasing trend and it was increased by four times from 1966 to 1981 [36]. The total inorganic P content entered into sea areas from Yangtze Estuary in 2002 was about 1.3 times that in the middle of 1980s [37]. Adequate nutrients input is in favor of algal growth, and thus the dominance of diatom is stable in the sea area [38].

The southern area in the ZCS is far away from the Yangtze Estuary and nutrients from the Yangtze Estuary to seas have been consumed significantly in the transport process, resulting in a sharp increase of N/P [39] and forming nutrients structural features of rich nitrogen and poor phosphorus [40,41,42]. Huang (2011) pointed that N/P continued to increase in the East China Sea and it climbed up from 8.5 in 1960s to 37 in 2010s [43]. The growth rate was increasing continuously. Phosphorus limit in the southern area in the ZCS was beneficial for the growth of dinoflagellate [44,45], which led to a decreased dominance of diatom.

In the eastern area, the dominance of diatom increased gradually before 1995, but it dropped significantly after 1995, showing obvious transition characteristics. This area might be a sensitive region to illustrate the variation of provenance.

### 4.2. Provenance Variation of BSi and TOC

According to analysis results on value of δ^13^C, there were temporal and spatial variations of provenance in the study area. The highest terrestrial input was in the northern area (A7-1). The mean δ^13^C was −23.46‰ at A7-1, but it declined sharply in 1970s, indicating the sudden increasing of terrestrial input in 1970s. Then, from 1970 to 1985, δ^13^C increased, indicating increasing of marine input during the period. However, the value of δ^13^C began to decrease continuously after 1985, indicating that terrestrial source input still had an increasing trend. In the southeastern (A10-4) and southern (A11-3) areas, marine source input occupied a dominant role. The mean δ^13^C was −22.53‰ in the southeastern area and −23.01‰ in the southern area. Specifically, the value of δ^13^C increased before 1995 and then declined in southeastern area, indicating that terrestrial-source input weakened firstly and then increased. In the southern area (A11-3), the numerical value of δ^13^C declined continuously, which reflected that terrestrial source input was increasing continuously. In recent years, terrestrial input in all the three areas had an increasing trend.

The intensified influences by terrestrial-source input in northern (A7-1) and southern (A11-3) coastal areas are widely believed in correlation with diluted water input from the Yangtze River [46]. The northern area is close to the Yangtze Estuary and diluted water from the Yangtze River carries excessive nutrients which are conducive to growth of algae [47]. The southern area is further away from the Yangtze Estuary and the structure of nutrients is high N/P which restricts the growth of diatom [48].

In the southeastern area (A10-4), the farthest offshore area, the dominance of diatom increased continuously before 1995, which reflected the increased marine input. It is speculated to be influenced by the upwelling which brings abundance phosphate. This caused a relatively low value of N/P, which is conducive to diatom growth [15,49]. Lou (2011) reported that upwelling could deliver bottom nutrients to the surface of sea as a supplementation to phosphate in southeastern area of Zhoushan Islands [50]. This provides a favorable condition for diatom growth. Hence, the variation in dominance of diatom in southeastern area may be related with changes of provenance which may be caused by upwelling.

## 5. Conclusions

In this study, the analysis results derived from the core data demonstrate that the primary productivity was higher in the northern area of Zhoushan Islands compared to that in the southern area, which suggests that the primary productivity is positively correlated to the distance from the Yangtze Estuary. This is because the Yangtze estuary inputs overload nutrient to the study area. However, according to the results TOC and BSi data analysis, there was an increasing trend in the northern and southern areas. There were also evident temporal and spatial changes in algae community structures in the study area. The algae community structure was stable in the northern area close to the Yangtze Estuary, and the dominance of diatom in the both southern and southeastern areas generally decreased. This might be closely related with source of nutrients. According to the results of δ^13^C, the source of nutrients in the northern area was controlled by terrestrial source input from Yangtze diluted water. In the southern area, δ^13^C also presented a decreasing trend, indicating the increasing terrestrial source input. However, in the southeastern area, marine source input took a dominant role during 1940–1995. It was speculated that the phosphates were supplemented in this area by upwelling, thus facilitating diatom growth. During 1996–2015, the value of δ^13^C was decreasing, indicating that the increasing terrestrial source input or the upwelling become weaker. This has restricted the growth of diatom and caused a low BSi/TOC since 1995.

To sum up, the eutrophication in the largest area of Zhoushan Coastal sea over the past 100 years was infected by terrestrial source input from the Yangtze diluted water. However, in the southeastern area, the role of marine source input from upwelling should be positive to enhance eutrophication in the study area. This should remind the managers of the coastal environment that we should not only control the overloading riverine input, but also pay more attention to marine nutrients input such as upwelling in this area.

## Figures and Tables

**Figure 1 ijerph-17-03890-f001:**
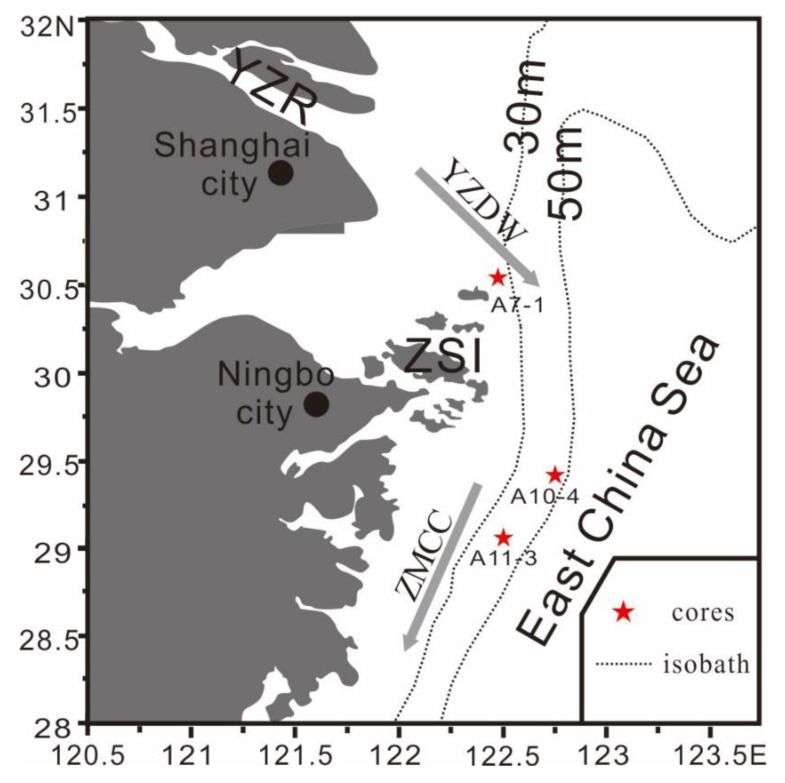
Study area and location of sampling (core samples) sites (YZR–Yangtze River, ZSI–Zhoushan Islands, YZDW–Yangtze Diluted Water, ZMCC–Zhe Min Coastal Current).

**Figure 2 ijerph-17-03890-f002:**
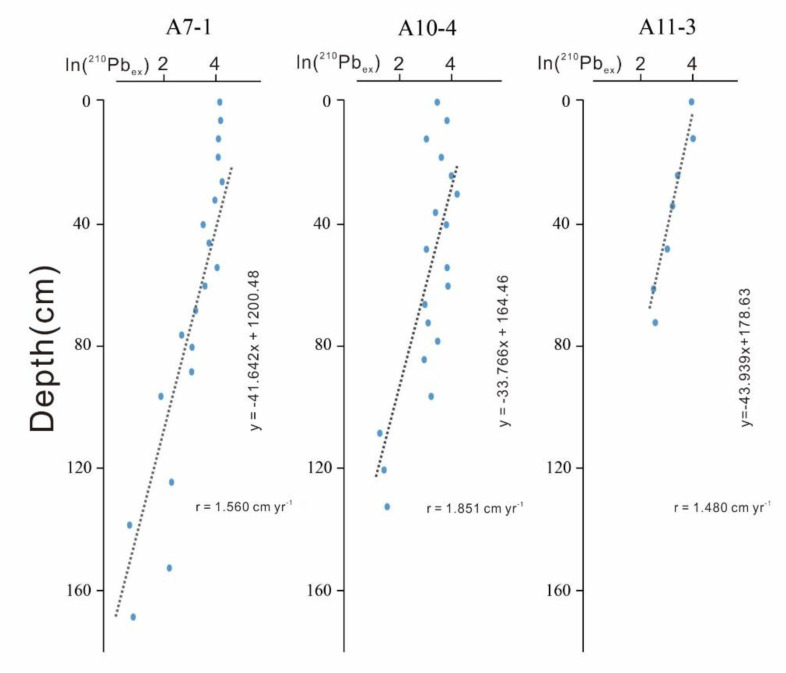
Results of ^210^Pb_ex._

**Figure 3 ijerph-17-03890-f003:**
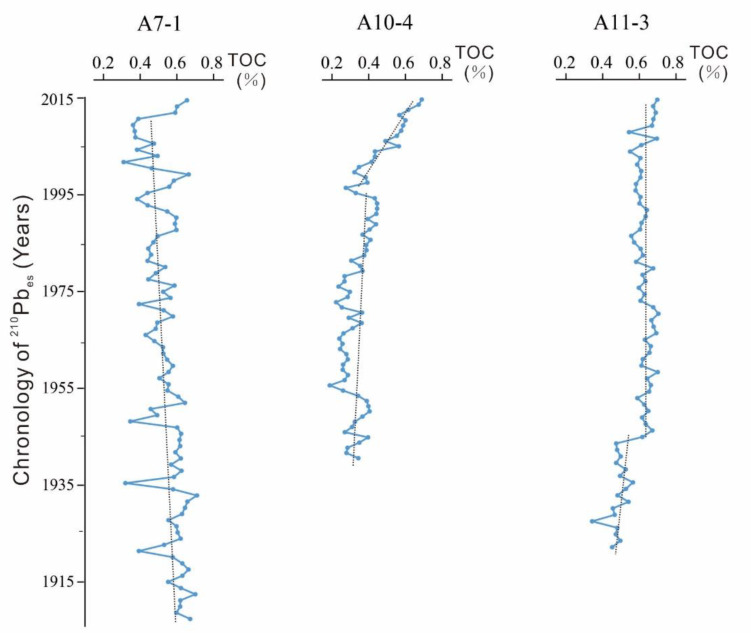
Results of TOC.

**Figure 4 ijerph-17-03890-f004:**
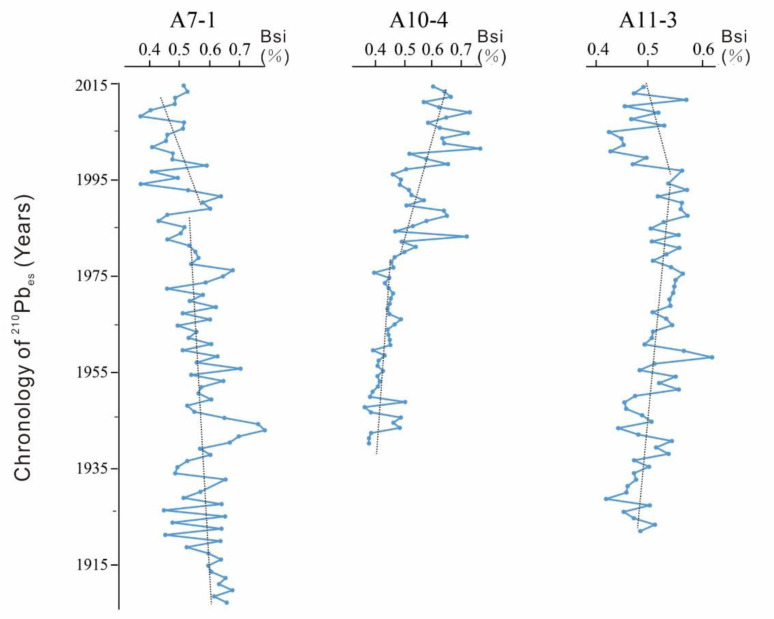
Results of BSi.

**Figure 5 ijerph-17-03890-f005:**
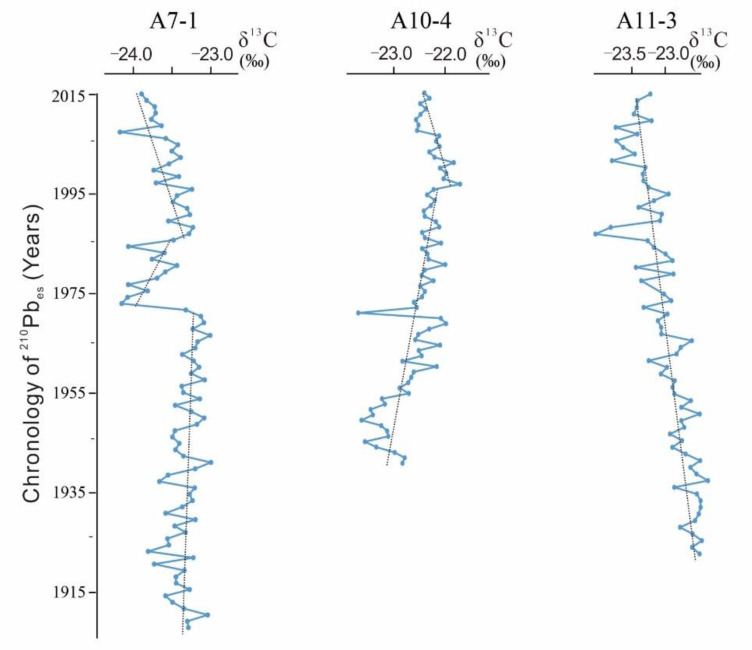
Results of the δ^13^C.

**Figure 6 ijerph-17-03890-f006:**
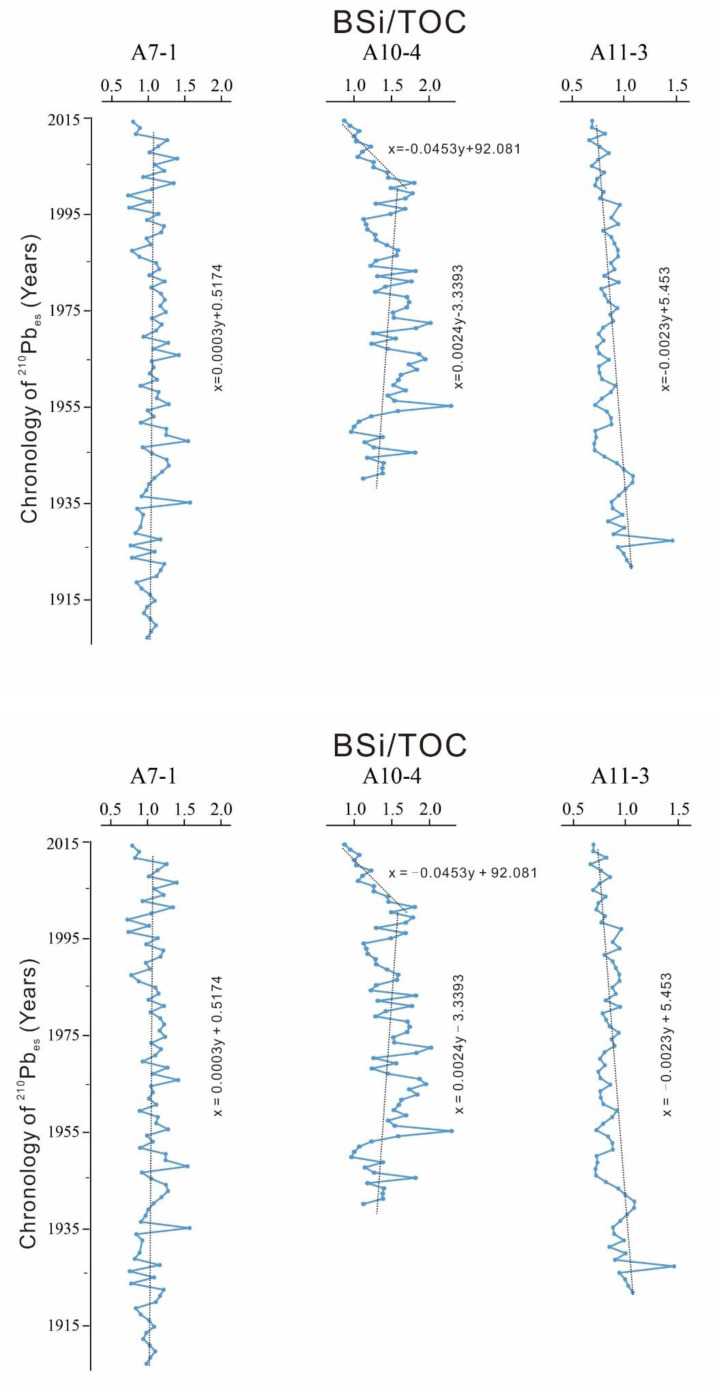
The variation of ratio of BSi/TOC.

**Table 1 ijerph-17-03890-t001:** Testing information.

Testing	A7-1	A10-4	A11-3
^210^Pb_ex_	19 samples (at the depth of 1 cm, 7 cm, 13 cm, 19 cm, 27 cm, 33 cm, 41 cm, 47 cm, 55 cm, 61 cm, 69 cm, 77 cm, 81 cm, 89 cm, 97 cm, 111 cm, 125 cm, 139 cm, 153 cm, and 169 cm)	19 samples (at the depth of 1 cm, 7 cm, 13 cm, 19 cm, 25 cm, 31 cm, 37 cm, 41 cm, 49 cm, 55 cm, 61 cm, 67 cm, 73 cm, 79 cm, 85 cm, 97 cm, 109 cm, 121 cm, 133 cm, and 141 cm)	7 samples (at the depth of 1 cm, 13 cm, 25 cm, 35 cm, 49 cm, 62 cm, 73 cm, and 85 cm)
BSi	85 samples (within the depth range 0~170 cm, at an interval of 2 cm)	70 samples (within the depth range 0~140 cm, at an interval of 2 cm)	70 samples (within the depth range 0~140 cm, at an interval of 2 cm)
TOC	85 samples (within the depth range 0~170 cm, at an interval of 2 cm)	70 samples (within the depth range 0~140 cm, at an interval of 2 cm)	70 samples (within the depth range 0~140 cm, at an interval of 2 cm)
δ^13^C	85 samples (within the depth range 0~170 cm, at an interval of 2 cm)	70 samples (within the depth range 0~140 cm, at an interval of 2 cm)	70 samples (within the depth range 0~140 cm, at an interval of 2 cm)

Note: ^210^Pb_ex_—the excess ^210^Pb; BSi—biogenic silica; TOC—total organic carbon; δ^13^C—stable carbon isotope.

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
