# Peer review of "Biogenic Silica and Organic Carbon Records in Zhoushan Coastal Sea over the Past One Hundred Years and Their Environmental Indications"

_ijerph, 2020, doi:10.3390/ijerph17113890_

Round 1
Reviewer 1 Report
Overall, although I am no expert on isotopic analyses this seems like an excellent manuscript (content, research design/methodology) that has a chance to provide a major contribution to the understanding of sources of excess nutrients that plague the Zhoushan Sea of China.
There are two big areas that need to be improved:
1) English language-most of the words (at least the roots of the words) are used correctly; however, the authors need to seek the assistance of a native english speaker to fix many verb tenses, sentence structures, and other issues that do not quite read correctly and take away from an otherwise excellent article. Here are two examples (but there are probably hundreds of grammatical problems in the article):
Lines 13-15: The Zhoushan coastal sea locates to the south of the Yangtze Estuary. Influenced by Yangtze diluted water, coastal current and upwelling, eutrophication in this area become serious recently, which is harmful to the health of coastal residents.
This part (first few lines of the abstract) needs to be significantly rewritten. For example:
The Zhoushan coastal sea is located south of the Yangtze Estuary. The Yangtze River and coastal currents and upwelling contribute to significant recent increases in eutrophication which threatens the health of coastal residents.
Line 291: that means the primary productivity has a positive correlate to the distance form Yangtze Estuary
This should read: this means that primary productivity is positively correlated to the distance from the Yangtze estuary
2) The authors do a nice job of relating algal blooms to public health threats in the introduction and need to do this again, more clearly in the conclusions--what are the public health and environmental policy implications of a greater understanding of the sources of nutrients?
Author Response
Overall, although I am no expert on isotopic analyses this seems like an excellent manuscript (content, research design/methodology) that has a chance to provide a major contribution to the understanding of sources of excess nutrients that plague the Zhoushan Sea of China.
Response: thanks for the positive comments on our manuscript.
There are two big areas that need to be improved:
1) English language-most of the words (at least the roots of the words) are used correctly; however, the authors need to seek the assistance of a native english speaker to fix many verb tenses, sentence structures, and other issues that do not quite read correctly and take away from an otherwise excellent article. Here are two examples (but there are probably hundreds of grammatical problems in the article):
Lines 13-15: The Zhoushan coastal sea locates to the south of the Yangtze Estuary. Influenced by Yangtze diluted water, coastal current and upwelling, eutrophication in this area become serious recently, which is harmful to the health of coastal residents.
This part (first few lines of the abstract) needs to be significantly rewritten. For example:
The Zhoushan coastal sea is located south of the Yangtze Estuary. The Yangtze River and coastal currents and upwelling contribute to significant recent increases in eutrophication which threatens the health of coastal residents.
Line 291: that means the primary productivity has a positive correlate to the distance form Yangtze Estuary
This should read: this means that primary productivity is positively correlated to the distance from the Yangtze estuary
Response: Thanks, we invited an English professional to revise and edit our manuscript throughout. All above problems have been addressed.
2) The authors do a nice job of relating algal blooms to public health threats in the introduction and need to do this again, more clearly in the conclusions--what are the public health and environmental policy implications of a greater understanding of the sources of nutrients?
Response: Thanks, the conclusions have been rewritten, and it became shorter and clearer.
Reviewer 2 Report
Lines 2-4 Title – isn’t legitimate : the methods, results, and conclusions of manuscript covers not only biogenic silica, but also others: TOC and C isotopic data. The environmental indications may be understood in many ways
Abstract –
Not comprehensible, it is not clear what did the Authors mean, writing:
southern area (line 36), northern area (line , southern Zoushan (line 23), north of Zhoushan (line 24), offshore area (lines 37-38) primary productivity.
Line 15 - Coastal residents – almost nothing of this subject in the text
Lines 37-38 - in the text marine input vs coastal input is little described
The aim is not well defined
Keywords – not all aprioppriate
Line 43 – please explain, what does it means: water environment
Line 78 Primary productivity
Line 78 Dominant species of algae
Line 79 Temporal and spatial variations – this is the main focus
Line 80 . in one core the age achieve 100 years , the others – less 100 years
Lines 89-90. abbreviations should be explained in the text
Fig 1. Legend YDZW , ZMCC - in the text nothing about that , seems not to be necessary
Table 1. nothing about the depths of samples, the legend (the abbreviations) is lacking
Line 96 Should be: Methods
Lines 97-103 poor described method: the parameters of measurements on the spectrometer is lacking; please put appropriate citations
Line 104. In the title the term: Biogenic silica” has been used, in the chapter 2.2.2. should be the same name.
Line 117. Chapter 2.2.3. Citation are lacking, the parameters of the measurements is lacking and also the parameters of equipment, which has been used
Line 126 Chap 3.1. too short and enigmatic! please explain details.
Line 132 Fig. 2 the results are not clear, the letters and signs in legend are too small, lack of the legend , error bars is lacking, plaes explain : e2, e 4; what we can say about values of 210 Pb ex
This chapter must be rewritten advantage
Line 149 Fig 3 – please, add the legend; should be discussed the error of the measurements
Lines 191-198 this part belong to discussion and interpretation, not to the results. The interpretation again is not clear: there are new factors: depth of cores and location due to the land, there weren’t mentioned before. The distance from the coast to the sampled cores is obviously important on the terrestrial material in the sediments component.
Lines 201 – 202 – It is result of this work?? If not - please put citation data in the text.
Lines 216-226 in the text there is not clear which core represent east area and which one the southern, again.
The growth rate in years are not visible on the figures, ; there were not pointed 1950 and 1986, as there are described in the text. Are there important data? Are these data known in the former literature ? Please add citation of earlier work, How this data are compatible with the data from The other numerous publications ( e.g. YANG et al, 2012) in column sediments?
Lines 239-241 – completely not understood part (TSS, GAM, Chlorophyll variation did not be mentioned before), please explain these abbreviations and their importance to this work
Line 241 [12] – Zhang in the bibliography, instead of Song (2014)
Line 241 Should be Zhou et al (2017), instead of Zhou (2017)
Line 242 The title is not clear: variation of provenance of what?
Lines 265-280 The northern and southern area is was not clear defined
Line 264 – variations of provenance of what?
Lines 269- please explain: in the east and southern areas
Lines 272-273 – the text: numerical value of delta C13 that terrestrial – source input was increasing continuously is not clear, give more explanation connected with the terrestrial – source input??
please discuss the results due to the errors of measurements and due to the earlier publications
Line 280 – please explain: in the east area,
Lines 283-287 – possibly but not documented in this work
Line 288 Conclusions – should be rewritten- we do not need the data, put only the results.
We do not find community structures in the text before. Please write it earlier in the text or throw it away from conclusions. Do not discuss in the conclusions data , speculative for this data N/P ratios and so on.
Please, show in the shorter form your essential conclusions resulted from this work!
Author Response
Lines 2-4 Title – isn’t legitimate : the methods, results, and conclusions of manuscript covers not only biogenic silica, but also others: TOC and C isotopic data. The environmental indications may be understood in many ways
Response: The title has been changed, the new title is “Biogenic Silica and Organic Carbon Records in Zhoushan Coastal Sea over the Past One Hundred Years and Theirs Environmental Indications”
Abstract –
Not comprehensible, it is not clear what did the Authors mean, writing:
southern area (line 36), northern area (line , southern Zoushan (line 23), north of Zhoushan (line 24), offshore area (lines 37-38) primary productivity.
Response: in the revised version, the abstract was rewritten.
Line 15 - Coastal residents – almost nothing of this subject in the text
Response: The part of “coastal residents” has been removed in the rewritten abstract.
Lines 37-38 - in the text marine input vs coastal input is little described
The aim is not well defined
Response: Thanks, we clarified the difference between them in the revised version.
Keywords – not all aprioppriate
Response: revised them and all were checked.
Line 43 – please explain, what does it means: water environment
Response: Thanks, “water environment” has been replaced by “Marine ecological environment” in the line 53 of new version.
Line 78 Primary productivity
Line 78 Dominant species of algae
Response: Thanks, we describe them clearer in the new version (line 92-93). We test the sedimentary record which can indicate the primary productivity and dominant species variation of algae over the 100 years.
Line 79 Temporal and spatial variations – this is the main focus
Response: Yes, thanks. We modified the end of instruction in the new version (line 95-97).
Line 80 . in one core the age achieve 100 years , the others – less 100 years
Response: The length of the three cores is all about 1.5 m, but the ages covered by the three cores are slightly different: the age of A7-1 is about 110 years; A11-3 is very close to 100 years, and A10-4 is about 75 years. Thus, we say the age with the three cores in this study is about 100 years.
Lines 89-90. abbreviations should be explained in the text
Response: Thanks, All abbreviations have been explained first, then their abbreviations were applied in the text.
Fig 1. Legend YDZW , ZMCC - in the text nothing about that , seems not to be necessary
Response: These two abbreviations, showing the flow directions in study area, help understanding the route of terrestrial input in the figure only.
Table 1. nothing about the depths of samples, the legend (the abbreviations) is lacking
Response: Thanks, this table has been modified, and the information of depths was added.
Line 96 Should be: Methods
Response: Done(line 119).
Lines 97-103 poor described method: the parameters of measurements on the spectrometer is lacking; please put appropriate citations
Response: we rewrote the part of the method. See Lines 121 – 136 (chapter 2.2.1) in the new editor version.
Line 104. In the title the term: Biogenic silica” has been used, in the chapter 2.2.2. should be the same name.
Response: Yes, thanks, changed (line 137).
Line 117. Chapter 2.2.3. Citation are lacking, the parameters of the measurements is lacking and also the parameters of equipment, which has been used
Response: Thanks, the method of TOC and d13C tests was added in detail, and 2 papers were cited (line 159-164).
Line 126 Chap 3.1. too short and enigmatic! please explain details.
Response: In the new version, detailed explanation was added(line 167-173).
Line 132 Fig. 2 the results are not clear, the letters and signs in legend are too small, lack of the legend , error bars is lacking, plaes explain : e2, e 4; what we can say about values of 210 Pb ex
This chapter must be rewritten advantage
Response: Thanks, Figure 2 has been replotted by adding the legend of x-axis, which is ln(210Pbex). The relevant part has been rewritten. We introduced the formula which cited from reference [29] and showed how the annual average deposition rate was calculated.
Line 149 Fig 3 – please, add the legend; should be discussed the error of the measurements
Response: Thanks, Figures 3 - 6 were improved.
Lines 191-198 this part belong to discussion and interpretation, not to the results. The interpretation again is not clear: there are new factors: depth of cores and location due to the land, there weren’t mentioned before. The distance from the coast to the sampled cores is obviously important on the terrestrial material in the sediments component.
Response: Thanks, this part has been merged into the discussion section. The depth of cores has been mentioned in “2.1. Collection of samples”(line 101-107)
Lines 201 – 202 – It is result of this work?? If not - please put citation data in the text.
Response: Thanks, in this revised version, we have added the citation info of the previous study about BSi/TOC ratio. This ratio can help illustrate the dominance of diatom, supported from existing work.
Lines 216-226 in the text there is not clear which core represent east area and which one the southern, again.
Response: Thanks, We added the core number for each specific area.
The growth rate in years are not visible on the figures, ; there were not pointed 1950 and 1986, as there are described in the text. Are there important data? Are these data known in the former literature ? Please add citation of earlier work, How this data are compatible with the data from The other numerous publications ( e.g. YANG et al, 2012) in column sediments?
Response: To respond the comments, in the new version we added a regression formula in Figure 6. We add the marks on y-axis. However, we couldn’t add every year on y-axis.
There are many existing studies on lots of BSi cores in Yangtze Estuary and East China Sea (e.g., Li summarized in 2016, cited as [28]). But only a few BSi data in the core sediment in this coastal sea are available. Discussion with the BSi data of Yangtze Estuary and East China Sea, associated with the nutrient, current, SST etc is out of the scope of the study.
In Lines 288-298, we cited numerous references, in which there were not column sediment BSi and TOC data but the observed algae data in the past 20 years or simulated modelling data. In this part, we did not discuss the trend of BSi/TOC ratio, but did discuss some influence factors that control the BSi/TOC variation.
Lines 239-241 – completely not understood part (TSS, GAM, Chlorophyll variation did not be mentioned before), please explain these abbreviations and their importance to this work
Response: Their full names have been added. They were explained in Lines 293 in the new version. This citation indicate that the influence of illumination on algae was small compare to nutrients in study area.
Line 241 [12] – Zhang in the bibliography, instead of Song (2014)
Response: Corrected.(line 295)
Line 241 Should be Zhou et al (2017), instead of Zhou (2017)
Response: Thanks, corrected. (line 295)
Line 242 The title is not clear: variation of provenance of what?
Response: Thanks, changed to “Provenance Variation of BSi and TOC”(line 319)
Lines 265-280 The northern and southern area is was not clear defined
Response: We added the core number in the new version to indicate which core represents which area. (line 270,273 and 278)
Line 264 – variations of provenance of what?
Response: Changed to “Provenance Variation of BSi and TOC” (line 319)
Lines 269- please explain: in the east and southern areas
Response: We added the core number for the two areas in the new version. (line 270,273 and 278)
Lines 272-273 – the text: numerical value of delta C13 that terrestrial – source input was increasing continuously is not clear, give more explanation connected with the terrestrial – source input??please discuss the results due to the errors of measurements and due to the earlier publications
Response: Thanks, we described the trend ofδ13C in A11-3. However, as we cited in the Methods, the tests were run by machines, and the errors of measurements were quite small. There were a few of earlier tests of BSi combining with TOC andδ13C available in the Zhoushan Coastal sea.
Line 280 – please explain: in the east area,
Response: Thanks, we added the explanation there.(line 326)
Lines 283-287 – possibly but not documented in this work
Response: Yes, this part was not based on our data, but cited from Lou (2011), the reference [50]. His study proved the low N/P caused by upwelling in this area, which might cause the dominance of diatom.
Line 288 Conclusions – should be rewritten- we do not need the data, put only the results.
We do not find community structures in the text before. Please write it earlier in the text or throw it away from conclusions. Do not discuss in the conclusions data , speculative for this data N/P ratios and so on.
Please, show in the shorter form your essential conclusions resulted from this work!
Response: Thanks, the conclusions have been rewritten, and the section becomes shorter and concise.
Round 2
Reviewer 2 Report
Dear Authors,
the manuscript have been corrected properly and and significantly improved. The conclusions were written correctly
Please note a few "lost" letters (e.g. in the lines 130 and 318).